# Evaluation of the Antihyperalgesic Potential of *Morus alba*, *Angelica archangelica*, *Valeriana officinalis*, and *Passiflora incarnata* in Alloxan-Induced Diabetic Neuropathy in Rats

**DOI:** 10.3390/cimb47090719

**Published:** 2025-09-04

**Authors:** Felicia Suciu, Ciprian Pușcașu, Dragos Paul Mihai, Anca Ungurianu, Corina Andrei, Robert Viorel Ancuceanu, Cerasela Elena Gîrd, Anne-Marie Ciobanu, Nicoleta Mirela Blebea, Violeta Popovici, Cristina Isabel Viorica Ghiță, Simona Negres

**Affiliations:** 1Faculty of Pharmacy, “Ovidius” University of Constanta, 900470 Constanta, Romania; felicia.suciu@drd.umfcd.ro (F.S.); nicoleta.blebea@365.univ-ovidius.ro (N.M.B.); 2Faculty of Pharmacy, “Carol Davila” University of Medicine and Pharmacy, 020956 Bucharest, Romania; dragos_mihai@umfcd.ro (D.P.M.); anca.ungurianu@umfcd.ro (A.U.); corina.andrei@umfcd.ro (C.A.); robert.ancuceanu@umfcd.ro (R.V.A.); cerasela.gird@umfcd.ro (C.E.G.); anne.ciobanu@umfcd.ro (A.-M.C.); simona.negres@umfcd.ro (S.N.); 3Center for Mountain Economics, “Costin C. Kritescu” National Institute of Economic Research (INCE-CEMONT), Romanian Academy, 725700 Vatra-Dornei, Romania; 4Faculty of Medicine, “Carol Davila” University of Medicine and Pharmacy, 050474 Bucharest, Romania; isabel.ghita@umfcd.ro

**Keywords:** diabetic neuropathy, medicinal plant extracts, antihyperalgesic activity, molecular docking, target prediction, AP2-associated protein kinase 1, pro-inflammatory cytokines

## Abstract

Diabetic neuropathy (DN) is one of the most prevalent complications of diabetes mellitus, affecting a substantial proportion of patients and contributing to progressive sensorimotor dysfunction. Despite its clinical significance, available treatments are often insufficient and associated with undesirable effects. This study aims to evaluate the potential of *Morus alba* (MA)*, Angelica archangelica* (AA)*, Valeriana officinalis* (VO)*,* and *Passiflora incarnata* (PI) extracts in ameliorating nociceptive alterations and inflammatory markers in the alloxan-induced diabetic rat model. Male Wistar rats with alloxan-induced DN received oral administration of the plant extracts (200 mg/kg/day) or gabapentin (100 mg/kg/day) for 15 days, the dosage regimen being established based on prior efficacy data in preclinical neuropathy models. Behavioral assessments of thermal and mechanical hypersensitivity were conducted using hot plate, tail withdrawal, von Frey, and Randall–Sellito tests. Tumor necrosis factor-α (TNF-α) and interleukin-6 (IL-6) levels were quantified in brain and liver homogenates to evaluate neuro-inflammatory responses. All plant extracts produced significant improvements in nociceptive thresholds compared to diabetic control, with the most marked effects observed for MA extract. Pro-inflammatory cytokine levels were significantly reduced in all treatment groups, with MA and AA extracts inducing the most significant reductions in TNF-α and IL-6 concentrations. Computational target prediction and molecular docking analyses revealed that key phytochemicals from the plant extracts may exert antihyperalgesic effects through multi-target modulation, notably via interactions with AAK1, a kinase involved in neuropathic pain signaling. The investigated plant extracts displayed significant antihyperalgesic and anti-inflammatory activities in a rat model of DN. Among them, MA extract revealed the most consistent therapeutic profile, supporting its potential role as a strategy for managing DN.

## 1. Introduction

Diabetic neuropathy (DN) is a prevalent and serious complication of both type 1 and type 2 diabetes mellitus, impacting up to half of patients worldwide [1]. Moreover, DN is associated with high morbidity, significantly diminished quality of life, and remains the leading cause of non-traumatic limb amputations [1,2,3].

Despite its clinical significance, the pathophysiology of DN is not fully understood, which contributes to the development of inadequate treatment strategies [4]. Multiple pathogenic mechanisms are involved, including oxidative stress, release of pro-inflammatory cytokines, activation of the polyol and hexosamine pathways, microvascular damage, and impaired insulin signaling [5].

Clinically, DN typically manifests as a symmetrical, length-dependent sensorimotor neuropathy, characterized by a stocking-glove distribution. Symptoms include both negative (numbness, reduced pain, and temperature perception) and positive (burning, tingling, electric shock-like sensations) sensory phenomena [6].

Currently, no approved drugs exist to improve the long-term prognosis of peripheral nerve damage, and an effective pharmacological treatment for neuropathic pain has yet to be discovered [7]. Standard first-line pharmacological therapies, such as tricyclic antidepressants, serotonin–norepinephrine reuptake inhibitors, and gabapentinoids [8], frequently offer only limited benefits and are associated with undesirable side effects [9,10,11,12,13,14,15]. Given these limitations, the development of new therapeutic approaches remains a priority.

Medicinal plants are increasingly investigated for their therapeutic potential in the treatment of chronic diseases, including diabetic complications. According to the World Health Organization, over 80% of the global population rely on herbal medicines. Their appeal lies in the presence of bioactive compounds that can modulate key biological pathways involved in disease pathophysiology, including inflammation and oxidative stress [16].

*Morus alba* L. (MA), also known as white mulberry, is the most prevalent species of the *Morus* genus. Rich in flavonoids (e.g., quercetin, kaempferol, and isorhamnetin), MA exhibits antioxidant, anti-inflammatory, hypoglycemic, and neuroprotective effects [17,18]. The bark of MA contains several active compounds, including morin (present at the highest concentration—12.3 µg/g) [19], albanol A [20], and sanggenon O [21], noted for its anti-inflammatory activity and modulation of gamma-aminobutyric acid type A (GABA_A_) receptors [22].

*Angelica archangelica* L. (AA), commonly known as garden angelica, has been traditionally used for nervous and digestive disorders. It contains essential oils and coumarins, particularly furanocoumarins, which are known for their analgesic, anti-inflammatory, and antioxidant properties [23,24,25,26]. In addition, furanocoumarin representatives, such as imperatorin, have been shown to activate the vanilloid receptor type 1 (TRPV1) [27], thereby underscoring its therapeutic potential in neuropathic pain.

*Passiflora incarnata* L. (PI), or passion flower, is used primarily for its sedative and anxiolytic effects. Its main constituents, vitexin and isovitexin, modulate gamma-aminobutyric acid type A and B (GABA_A_ and GABA_B_) receptors and gamma-aminobutyric acid (GABA) uptake. While limited, some evidence suggests potential antinociceptive effects in neuropathy models, possibly via GABAergic and opioid mechanisms [28,29]. Moreover, Aman et al. highlighted the impact of reducing diabetic neuropathy allodynia in a streptozotocin-induced model of diabetes [30].

*Valeriana officinalis* L. (VO), also known as “All-heal,” has long been valued for its calming effects. Neurobiological research showed that valerenic acid interacts with the GABAergic system, a mechanism of action similar to that of benzodiazepine drugs [31,32]. VO also demonstrated antioxidant and anti-inflammatory activities, which may reduce neuroinflammation associated with neuropathic pain [33,34,35]. Previous studies have shown that valerian extract exerted analgesic activities in models of fibromyalgia or orofacial pain induced by the administration of 2.5% formaldehyde.

These four species were selected based on their phytochemical profiles, particularly their capacity to modulate GABAergic signaling—a pathway closely associated with nociceptive regulation. Neuropathic pain involves disrupted GABA synthesis, downregulation of GAD (glutamic acid decarboxylase) and GABA transporters, impaired chloride transporter function (NKCC1/KCC2), and altered GABA receptor activity, all of which contribute to neuronal hyperexcitability and central sensitization [31]. Gabapentin, a GABA analog, remains a first-line agent for DN, further supporting the therapeutic relevance of this pathway [36,37,38]. All four medicinal plant extracts share a functional ability to modulate GABA-mediated pathways, providing a strong neurochemical rationale for their investigation in neuropathic pain. Quantitative analyses have confirmed the presence of GABA in various *Morus* species, with MA containing approximately 0.640% GABA, as reported by Chen et al. [39]. In PI, the whole plant extract has shown potent modulatory effects on GABA_A_ receptor activity, attributed to its elevated GABA content [40]. In AA, compounds such as columbianetin, imperatorin, cnidilin, osthol, and columbianedin have been found to enhance GABA-induced chloride influx via GABA_A_ receptors in a dose-dependent manner [41]. Similarly, VO and its primary active compound, valerenic acid, are known to modulate GABA_A_ receptor function, potentially enhancing the effects of GABAergic drugs and anesthetics [42]. Taken together, the GABAergic, antioxidant, and anti-inflammatory properties of these four species support their potential as therapeutic agents for the treatment of DN.

The present study aims to evaluate the antihyperalgesic potential of the hydro-ethanolic extracts of these four medicinal plants (MAE, AAE, VOE, and PIE) through a complex approach, involving four experimental in vivo assays on rats with alloxan-induced diabetes (hot plate, tail withdrawal, von Frey, and Randall–Sellito), biochemical investigations of inflammatory markers, and in silico analyses. Our experimental results, supported by an extensive statistical analysis, provide a comprehensive assessment of the therapeutic potential of these medicinal plants in DN management, as well as insight into the molecular mechanisms underlying their antihyperalgesic effects.

## 2. Materials and Methods

### 2.1. Plant Material and Extracts

Dried roots of AA and VO (SC Stef Mar SRL, Râmnicu Vâlcea, Romania) and aerial parts of PI (Laboratoarele Fares Bio Vital SRL, Orăștie, Romania), were purchased from commercial retailers. The bark of MA was collected in May 2018 from Buzău County, Romania. The MA bark was initially air-dried at ambient temperature, followed by oven drying at 55 °C for several days until it became brittle and suitable for grinding.

The plant materials were milled using a Swantech electric grinder (Swantech Ltd, Selby, UK), and the resulting powder was sieved through a V-type mesh (250 µm). Extraction was performed twice using ethanol: 70% (*v*/*v*) for MA bark and 50% (*v*/*v*) for AA and VO roots and PI aerial parts, applying hot reflux at approximately 75 °C. The different ethanol concentrations were chosen to optimize yield for each plant matrix. A higher hydro-ethanolic solution (70%) efficiently solubilizes the abundant polyphenols from MA bark. In contrast, 50% ethanol effectively extracts the bioactive coumarins and flavonoids of AA, VO, and PI, limiting the co-extraction of impurities. The extracts (MAE, AAE, VOE, and PIE) were filtered through filter paper, and the combined filtrates were concentrated under reduced pressure using a rotary evaporator (BÜCHI Labortechnik AG, Flawil, Switzerland) at 70 °C. Then, they were freeze-dried at –58 °C (shelf temperature) using a ScanVac CoolSafe lyophilizer (LaboGene A/S, Lillerød, Denmark) to preserve heat-sensitive phytochemicals and ensure complete solvent removal [43].

### 2.2. Experimental Animals

All procedures involving animals complied with the ethical standards outlined in Romanian Law No. 43/2014 on the protection of animals used for scientific purposes and were conducted in accordance with Directive 2010/63/EU of the European Parliament. The study protocol was approved by the Bioethics Committee of the Faculty of Pharmacy, “Carol Davila” University of Medicine and Pharmacy, Bucharest, Romania (Approval No. 11/08 July 2022).

A total of eighty-three male Wistar rats, aged between 8 and 10 weeks and weighing 300 ± 10 g, were obtained from the “Cantacuzino” National Institute of Research and Development for Microbiology and Immunology, Bucharest, Romania. The animals were kept in standard plexiglass cages (Geplast SRL, Constanta, Romania) under controlled environmental conditions. They had free access to drinking water and a standard rodent pellet diet provided by the supplier of the laboratory animals. The housing conditions consisted of a temperature of 21–24 °C and a relative humidity of 45–60%, as monitored by a hygrothermal system. Before the experiment began, the rats underwent a one-week acclimatization period.

### 2.3. Diabetes Mellitus Induction and Applied Treatments

Diabetes was induced in rats by a single intraperitoneal (i.p.) injection of alloxan (Sigma Aldrich, Hamburg, Germany; product no. A7413) at a dose of 130 mg/kg, after a 24 h fasting period. Based on previous studies, which reported an approximately 66% success rate in inducing diabetes with alloxan, a total of 75 rats were used for this procedure [44]. Forty-eight hours after administration, blood glucose levels were measured using an ACCU-CHEK Active glucometer (Roche Diagnostics, Penzberg, Germany) by tail vein puncture [44]. Of the 75 rats, 48 developed hyperglycemia (blood glucose level > 180 mg/dL) and were randomly assigned to six experimental groups (*n* = 8 per group). Additionally, eight healthy rats were included as a non-diabetic control group.

The substances were administered orally once daily for 15 consecutive days. The experimental groups were established as follows: (i) ND (non-diabetic control) and D (diabetic control) received distilled water at a dose of 1 mL/kg. (ii) The GBN group was treated with gabapentin (Egis Pharmaceuticals PLC, Budapest, Hungary) in a water solution, at a dose of 100 mg/kg. (iii) Four distinct groups were treated with plant extracts (VOE, PIE, AAE, and MAE). These extracts were administered as solutions or suspensions, prepared by dissolving each dry extract in distilled water, and stored in a refrigerator at 2–8 °C, with use within a maximum of two days.

The plant extract dose of 200 mg/kg was chosen based on prior studies that investigated its antinociceptive activity. Thus, the MA bark extract dose was selected based on preclinical data indicating significant analgesic effects at this level, including reduced writhing responses and increased latency in thermal pain assays [45]. *Valeriana officinalis* has a long-standing history of traditional use for nervous system disorders; however, experimental evidence supporting its antinociceptive activity remains limited. One preclinical study reported significant effects at 200 and 800 mg/kg, reflected in increased tail-flick latency and reduced writhing. Given our focus on safety and translational relevance, a dose of 200 mg/kg was selected as a conservative yet pharmacologically active option, aiming to reduce the risk of central nervous system depression while preserving efficacy [46]. For AAE, the selected 200 mg/kg dose was supported by a murine fibromyalgia model, in which oral administration of an ethanolic extract led to significant reductions in pain behaviors. These findings suggest broader analgesic effects beyond a specific pain model [47]. In the case of PIE, our choice was based on a streptozotocin-induced DN study, where oral administration of the ethanolic extract at 200 and 300 mg/kg resulted in significant anti-allodynic effects [30]. The GBN dose of 100 mg/kg is effective, as indicated by previous DN studies [48,49].

### 2.4. Blood Glucose Levels

Blood glucose levels (expressed in mg/dL) were evaluated three times: initially, on day 7, and on day 15, following group allocation. The blood samples for analysis were obtained via tail vein puncture.

### 2.5. Experimental Assays for Evaluating Antihyperalgesic Effects

#### 2.5.1. Heat Hypersensitivity

Thermal hyperalgesia was assessed using the hot plate test, performed at baseline and on days 7 and 14 of treatment. The rats were exposed to a thermal stimulus by being placed on a heated surface maintained at 53 °C. The latency time, defined as the interval between placement on the hot plate and the first nocifensive response, such as paw licking, shaking, or jumping, was recorded as an indicator of pain sensitivity. To prevent tissue damage, a cutoff time of 25 s was applied [50].

#### 2.5.2. Cold Hypersensitivity

Cold sensitivity was evaluated using the tail withdrawal test. For this, each rat’s tail was submerged halfway into water maintained at 10 ± 0.5 °C utilizing an ice bath, and the latency until tail withdrawal was recorded. To prevent tissue damage, a cutoff time of 25 s was applied. Measurements were performed at baseline and on days 7 and 14 after substance administration [51].

#### 2.5.3. Tactile Hypersensitivity

Tactile hypersensitivity was evaluated using von Frey filaments (Ugo Basile, Varese, Italy) at baseline, as well as on days 8 and 15 of treatment. Before testing, the animals were placed in individual plexiglass enclosures situated on a perforated wire mesh platform and allowed to acclimate for 30 min. A series of calibrated von Frey filaments with increasing bending forces (1.4, 2, 4, 6, 8, 10, 15, and 26 g), corresponding to sizes 4.17 to 5.46, were applied perpendicularly to the plantar surface of both hind paws. Each filament was pressed with enough force to cause a slight bend and held in place for 6 s. The testing began with filament number 4 (applying a force of 6 g). If the animal did not withdraw its paw, the next filament with greater stiffness was applied, and the response was marked as negative (O). If the animal withdrew its paw, this was considered a positive response and marked as (X), followed by the use of the next filament with lower stiffness. Once a crossover pattern (OX or XO) was observed, or after four consecutive identical responses, the trial was concluded [52]. The 50% paw withdrawal threshold was calculated using Dixon’s up-and-down method [53], as validated by Chaplan et al. [54].

#### 2.5.4. Mechanical Hyperalgesia

The Randall–Selitto test was employed to evaluate the pressure-induced hyperalgesia by applying a gradually increasing force to the right hind paw using an analgesy meter (Ugo Basile, model 37215, Verase, Italy). The assessments were conducted at baseline, as well as on days 7 and 14 of treatment. The pressure was applied until the animal exhibited a vocalization response or the maximum threshold of 250 g was reached, whichever occurred first [55].

### 2.6. Biochemical Analysis of Brain and Liver Homogenates of Rats

The rats were sacrificed by receiving an intraperitoneal (i.p.) dose of 200 mg/kg thiopental sodium (Sigma Aldrich, St. Louis, MO, USA) [56]. Then, the brain and liver tissues were prelevated for further analysis. The tissue homogenates were prepared by mixing the organs with phosphate-buffered saline (PBS) in a 1:10 (*w*/*v*) ratio using an RW 14 basic homogenizer (IKA, Staufen, Germany). The resulting homogenates were then diluted 1:10 with PBS before undergoing experimental procedures.

#### Evaluation of TNF-α (Tumor Necrosis Factor-α and IL-6 (Interleukin-6) Levels

The levels of TNF-α (Catalog No. BMS622) and IL-6 (Catalog No. BMS625) were measured according to the manufacturer’s instructions (Invitrogen, Thermo Fisher Scientific, Waltham, MA, USA).

### 2.7. Statistical Analysis

Statistical analysis was carried out using GraphPad Prism software, version 5.00 (GraphPad Software, San Diego, CA, USA). Data normality was evaluated using the D’Agostino–Pearson test. For datasets that followed a normal distribution, one-way ANOVA was employed, followed by Dunnett’s multiple comparisons test. When the data were not normally distributed, the Kruskal–Wallis test was applied, accompanied by Dunn’s post hoc analysis. A *p*-value < 0.05 was considered statistically significant. The results are expressed as mean ± standard error of the mean (SEM). Percentage differences between experimental groups were determined using Formula (1) as described in reference [50]. As fold changes are applicable, for reference, a “fold change” denotes the ratio of a treatment group’s mean to that of the control (e.g., a 2-fold increase indicates the mean value is twice that of the control).∆% = (M_x_ − M_y_)/M_y_ × 100(1)
where M_x_ represents the mean value of the D group when compared to the ND group, or the mean value of the GPN, AAE, PIE, MAE, and VOE groups when compared to the D group; M_y_ denotes the mean value of either the ND or D group, as appropriate.

### 2.8. Computational Methods

#### 2.8.1. Molecular Target Prediction

Potential molecular targets of the phytochemicals previously identified in the investigated plant extracts by UHPLC-HRMS/MS in our earlier study [57] were predicted using the Super-PRED online platform (https://prediction.charite.de, accessed on 10 April 2025). This computational tool integrates machine-learning algorithms and chemical structure similarity analysis to infer potential protein targets based on known bioactivity data. From the predicted target list, we selected those specifically associated with indications labeled as “diabetic neuropathy”, “peripheral neuropathy”, and “neuropathic pain” for further evaluation. These selected targets were subsequently subjected to molecular docking studies to assess binding affinities and investigate possible molecular interactions between the chosen phytochemicals and their predicted protein targets.

#### 2.8.2. Molecular Docking Simulations

Molecular docking analyses were conducted to investigate the potential binding interactions between phytochemicals previously identified by UHPLC-HRMS/MS and AP2-associated protein kinase 1 (AAK1), a molecular target predicted by the SuperPred web server (https://prediction.charite.de/index.php, accessed on 10 May 2025). The crystal structure selected for docking simulations was retrieved from the RCSB Protein Data Bank (PDB ID: 4WSQ, 1.95 Å resolution [58]), co-crystallized with the kinase inhibitor K-252A (https://www.rcsb.org/, accessed on 10 May 2025).

Preparation of the protein structure was conducted using YASARA v. 25.1.13 Structure software [59]. This process included the removal of co-crystallized ligands and water molecules, correction of structural inconsistencies, protonation of amino acid side chains at physiological pH (7.4), and optimization of hydrogen bonding networks. Subsequently, energy minimization was performed utilizing the YASARA2 force field. The docking protocol was validated by re-docking the original ligand, K-252A, into the ATP-binding site. The accuracy of docking was assessed by computing the root mean square deviation (RMSD) between the predicted pose of the co-crystallized ligand and the experimentally determined crystallographic conformation.

Phytochemical compounds were prepared by obtaining their SMILES representations from the PubChem database (https://pubchem.ncbi.nlm.nih.gov/, accessed on 10 May 2025) [60]. Their 3D molecular structures were subsequently generated and protonated at physiological pH using DataWarrior v5.2.1 [61], followed by energy minimization employing the MMFF94s+ force field.

Docking simulations were performed using AutoDock Vina v1.1.2 [62], targeting the ATP-binding pocket originally occupied by K-252A in the crystal structure. Each phytochemical underwent twelve independent docking runs. Predicted binding energies (ΔG, kcal/mol) and ligand efficiency (LE, defined as ΔG per heavy atom) were recorded. Visualization and further analysis of the ligand–protein interactions were conducted using BIOVIA Discovery Studio Visualizer software (Version 17.2.0, Dassault Systèmes, 2016, San Diego, CA, USA).

## 3. Results

### 3.1. Blood Glucose Levels

Following the induction of diabetes using alloxan, blood glucose levels showed notable differences between the ND and the D groups. One-way ANOVA revealed a statistically significant difference between treatment groups (F = 19.59, *p*-value < 0.0001, Figure 1A). All diabetic rats exhibited significantly increased blood glucose levels compared to the ND group (*p* < 0.001, Dunnett correction; Figure 1A).

After 7 days of treatment, all diabetic rats continued to exhibit significantly elevated blood glucose levels compared to the ND (Kruskal–Wallis, H = 22.41, *p* = 0.0010, Figure 1B). By the end of the experiment, blood glucose levels remained significantly higher in all treated groups relative to the ND group (univariate ANOVA, F = 36.08, *p* < 0.0001, Figure 1C). However, all four plant extracts led to a significant reduction in glucose levels compared to the D group. The blood glucose levels remained elevated; none of the diabetic rats achieved normoglycemia (*p* < 0.001, Dunn’s post hoc test, Figure 1C). The most notable decrease was observed in rats with MAE, showing a 45.7% reduction compared to the D group.

### 3.2. Experimental Tests for Evaluating Antihyperalgesic Effects

#### 3.2.1. Heat Hypersensitivity

Our study revealed significant differences in pain response latency in the hot plate test between the ND group and all diabetic groups before drug administration (univariate ANOVA, F = 4.938, *p* = 0.0005, Figure 2A). Diabetic rats showed markedly lower pain response values compared to the ND group (*p* < 0.001, Dunnett correction). Significant alterations in thermal pain sensitivity were observed following seven consecutive days of treatment (Kruskal–Wallis, H = 19.55, *p* = 0.0033; Figure 2B). Further pronounced differences were detected after 14 days (Kruskal–Wallis, H = 26.54, *p* = 0.0002; Figure 2C). The GBN group exhibited a notable increase in pain response latency at both 7 and 14 days compared to the D group, showing the most substantial antihyperalgesic effect among all treatment groups at day 14 (a 3.05-fold increase, 205% relative to D, *p* < 0.001, Dunn correction; Figure 2C). Among the diabetic rats receiving plant extracts, all demonstrated significantly prolonged pain response latency after 7 days of treatment compared to the D group, with effects comparable to those of GBN (*p* < 0.01, Dunn correction; Figure 2B). At the 14-day mark, all rats treated with plant extracts showed significantly increased latency values compared to the diabetic control; the most substantial improvement was obtained with MAE (2.88-fold increase, 188% vs. D), followed closely by the VOE (2.84-fold increase) (*p* < 0.05, Dunn correction; Figure 2C).

#### 3.2.2. Cold Hypersensitivity

Before treatment, significant differences in cold-induced pain sensitivity were observed between the non-diabetic (ND) group and all diabetic groups (univariate ANOVA, F = 4.357, *p* = 0.0013; Figure 2D). In the tail withdrawal test performed with water cooled to 10 °C, all diabetic groups exhibited significantly heightened sensitivity to cold stimuli compared to the ND group at baseline (*p* < 0.01, Dunnett correction; Figure 2D). Notable changes in pain response latency were observed after 7 days (univariate ANOVA, F = 12.54, *p* < 0.0001, Figure 2E) and became even more pronounced after 14 days of treatment (Kruskal–Wallis, H = 27.69, *p* < 0.0001; Figure 2F). Gabapentin-treated rats showed a marked improvement in latency times, with a 3.23-fold increase at 7 days (223%) and a 3.39-fold increase (239%) at 14 days compared to the D group (*p* < 0.001 for both time points, Dunnett and Dunn corrections; Figure 2E,F). Similarly, all diabetic rats treated with plant extracts displayed significantly reduced pain sensitivity at both time points compared to untreated diabetic controls (*p* < 0.05). After 14 days, the effects mirrored those seen in the heat hypersensitivity test, with MA extract exhibiting the most prominent improvement, comparable to gabapentin, with a 3.37-fold increase (237%, *p* < 0.001, Dunn correction), followed closely by VO extract, which produced a 3.33-fold increase (233%, *p* < 0.001).

#### 3.2.3. Tactile Hypersensitivity

Following diabetes induction, an initial assessment was conducted before therapy initiation, revealing significant intergroup differences (Kruskal–Wallis, H = 19.62, *p* = 0.0032; Figure 3A). Diabetic groups exhibited a marked increase in pain sensitivity compared to the ND group (*p* < 0.05, Dunn correction; Figure 3A). After 8 days of treatment, notable changes in tactile sensitivity were observed (Kruskal–Wallis, H = 24.38, *p* = 0.0004; Figure 3B), which persisted throughout the experimental period (univariate ANOVA, F = 21.17, *p* < 0.0001; Figure 3C). All treated groups showed a reduction in pain sensitivity after 8 days; however, only the rats treated with GBN, AAE, PIE, and VOE showed statistically significant improvements compared to the untreated diabetic group (*p* < 0.05, Dunnett correction; Figure 3B). The effects of GBN, PIE, and VOE were comparable. After 15 days of treatment, all diabetic groups receiving treatment showed significant increases in the 50% paw withdrawal threshold in the von Frey test compared to the D group (*p* < 0.001, Figure 3C). The most substantial antihyperalgesic effect was observed in the GBN-treated rats, which demonstrated a 3.27-fold increase (237%) over the diabetic control. Among the plant extract-treated animals, the most pronounced effect was observed with MAE, resulting in a 2.88-fold increase (188%), followed by AAE with a 2.74-fold increase (174%) compared to the D group.

#### 3.2.4. Mechanical Hyperalgesia

In the Randall–Sellito test, significant differences in mechanical hyperalgesia were observed between the ND control group and all diabetic groups before treatment initiation (univariate ANOVA, F = 18.33, *p* < 0.001; Figure 3D). All diabetic rats exhibited a significantly reduced pain threshold compared to the ND group (*p* < 0.001; Figure 3D). After seven consecutive days of treatment, mechanical sensitivity differed significantly among the groups (Kruskal–Wallis test, H = 19.55, *p* = 0.0033; Figure 3E). A similar pattern was observed after 14 days of treatment (univariate ANOVA, F = 8.231, *p* < 0.0001; Figure 3F). At day 7, all treated diabetic rats exhibited a reduction in mechanical pain sensitivity; however, the effect was statistically significant only for the GBN, PIE, and VOE-treated groups (*p* < 0.01, Dunn’s correction; Figure 3E). These animals displayed comparable levels of antihyperalgesic effect. By the end of the experiment, results were consistent with those obtained in the von Frey test. The GBN group demonstrated the most significant increase in maximum tolerated force, with an improvement i 88.84% compared to the D group (*p* < 0.0001, Dunnett correction; Figure 3F). Among the plant extract-treated rats, MAE revealed the most substantial effect (66.17%), followed by AAE (58.33%) (*p* < 0.0001, Dunnett correction; Figure 3F).

### 3.3. Biochemical Analysis of Brain and Liver Homogenates

#### Evaluation of TNF-α and IL-6 Levels

Considering the cytotoxic impact of alloxan-induced diabetes on various organs, such as the brain, pancreas, liver, and kidneys [63], we analyzed TNF-α and IL-6 levels in brain and liver tissues at the end of the experiment.

Regarding the TNF-α levels, the ANOVA test revealed significant differences among the groups in both brain tissues (F = 46.49, *p* < 0.001; Figure 4A) and liver tissues (F = 51.30, *p* < 0.001; Figure 4B). TNF-α concentrations were significantly elevated in the D group compared to the ND group. Moreover, all treated diabetic rats showed a significant reduction in TNF-α levels relative to the D group. In brain tissue, the AAE produced the most substantial decrease (64.25%), followed by the MAE (61.64%) and GBN (60.64%) (*p* < 0.001, Dunnett correction; Figure 4A). Similarly, in liver tissue, while the D group exhibited a marked increase in TNF-α levels, treatment with all other agents led to significant reductions (*p* < 0.001, Dunnett correction; Figure 4B). The MA extract promoted the most pronounced effect (71.4% reduction compared to the D group). Biochemical analysis of brain (ANOVA, F = 15.89, *p* < 0.001; Figure 4C) and liver (ANOVA, F = 47.59, *p* < 0.001; Figure 4D) tissues revealed significant changes in IL-6 levels following 15 treatment days. In both organs, IL-6 concentrations were markedly higher in the diabetic group compared to the ND (*p* < 0.001, Dunnett post hoc test; Figure 4C,D). In brain tissue, the treatment with GBN and all four plant extracts resulted in a significant reduction in IL-6 levels compared to the D group (*p* < 0.001, Dunnett post hoc test; Figure 4C). The AAE, MAE, and GBN induced comparable reductions in IL-6 levels, with decreases of 54.34%, 53.99%, and 49.32%, respectively. Similarly, in liver tissue, the administration of GBN and plant extracts significantly decreased IL-6 concentrations compared to the D group (*p* < 0.001, Dunnett post hoc test; Figure 4D). The most pronounced reduction was obtained with the MAE treatment (60.18%), followed closely by GBN (59.45%) and the AAE (58.77%).

### 3.4. Computational Studies

#### 3.4.1. Prediction of Neuropathy-Associated Targets

To uncover the potential molecular mechanisms underlying the observed antihyperalgesic effects of the plant extracts in the alloxan-induced diabetic neuropathy, target prediction was performed using the Super-PRED online server. This computational approach identified several phytochemicals with a significant likelihood of modulating molecular targets previously linked to diabetic neuropathy, peripheral neuropathy, and neuropathic pain (Table 1).

Multiple flavonoids were predicted to interact with the voltage-gated N-type calcium channel (CaV2.2) and glutamate ionotropic receptor AMPA type subunit 2 (GluA2). For instance, formononetin (0.7292), hesperetin (0.8026), hyperoside (0.5644), pinocembrin (0.6498), and pinostrobin (0.7959) showed substantial probabilities for interactions with CaV2.2, which is critically involved in pain signaling pathways, and its inhibition has been established as an effective strategy for managing neuropathic pain [64].

Compounds such as genistin (0.9175), glycitein (0.7938), hyperoside (0.8577), and kaempferol (0.6827) were predicted to interact with GluA2, which was previously shown to be involved in the modulation of synaptic plasticity and excitatory neurotransmission, processes closely associated with neuropathic pain and peripheral sensitization [65].

The adaptor-associated kinase 1 (AAK1), a serine/threonine protein kinase, was another relevant target predicted for several compounds, including apigenin (0.7149), catechin (0.6417), daidzein (0.6016), genistein (0.6801), hesperetin (0.5834), and naringenin (0.738). AAK1 is involved in intracellular signaling pathways that regulate nociceptive transmission and neuronal excitability, highlighting its potential as a therapeutic target for neuropathic pain management.

Aldose reductase (AR) emerged as a noteworthy target, particularly due to its pivotal role in diabetic neuropathy. AR is a key enzyme in the polyol pathway, responsible for converting glucose into sorbitol. Under hyperglycemic conditions, excessive activation of AR leads to accumulation of sorbitol, osmotic stress, oxidative damage, and nerve dysfunction, making AR inhibition a therapeutic strategy in diabetic neuropathy [66]. Compounds predicted to interact with AR, such as isorhamnetin (0.6451), kaempferol (0.5024), and rutin (0.6648), could therefore exert beneficial effects by reducing sorbitol accumulation and oxidative stress.

Additional predicted interactions included the muscarinic acetylcholine receptor M_1_ (M1R) with compounds such as chlorogenic acid (0.8814), cinnamic acid (0.6955), and galangin (0.649). M1R modulation is recognized for influencing pain perception and inflammatory responses [67].

These predicted molecular interactions suggest that the identified phytochemicals may contribute significantly to the observed antihyperalgesic effects of the studied plant extracts. Consequently, AP2-associated protein kinase 1 (AAK1), identified as a relevant molecular target, was selected for subsequent molecular docking analyses to explore binding interactions at the molecular level further.

#### 3.4.2. Molecular Docking

Molecular docking studies were conducted to assess the binding affinities and potential interactions between selected phytochemicals and AAK1. The docking protocol was validated by re-docking the original ligand, K-252A, into the active site of AAK1 (PDB ID: 4WSQ). The validation procedure yielded a root mean square deviation (RMSD) value of 0.9281 Å (Figure 5), confirming the reliability and accuracy of the pose prediction (RMSD < 2.0 Å). Moreover, the predicted binding energy of the co-crystallized ligand was −13.503 kcal/mol. Conversely, phenolic acids such as caffeic acid (–6.581 kcal/mol), ferulic acid (–6.546 kcal/mol), *p*-coumaric acid (–6.329 kcal/mol), and cinnamic acid (–6.327 kcal/mol) displayed comparatively lower affinities. Still, they demonstrated notably high ligand efficiencies (ranging from 0.4676 to 0.5752), highlighting their efficient binding relative to their smaller size (Table 2).

After visual inspection of the predicted docked pose, we selected apigenin and pinostrobin for further evaluation. Docking analysis of apigenin within the ATP-binding cleft of AAK1 revealed a well-defined network of stabilizing interactions involving both polar and hydrophobic contacts. A key conventional hydrogen bond was formed with Cys129, a residue located in the hinge region that plays a critical role in anchoring ATP-competitive kinase inhibitors (Figure 6). Two additional hydrogen bonds were observed: one with Asp127, part of the loop preceding the DFG motif, and another with Gly55, situated in the β1 strand forming the base of the nucleotide-binding pocket. These interactions likely contribute to the precise orientation of the ligand. Hydrophobic interactions further reinforced the binding of apigenin. Residues Leu52 and Val60, components of the glycine-rich loop (P-loop), established π–sigma and π–alkyl contacts with the chromone core of apigenin, stabilizing its position near the phosphate-binding region. Residue Leu183, located adjacent to the αC helix, contributed two π–sigma interactions, enhancing hydrophobic packing. Notably, a π-sulfur interaction was observed with Met126, positioned near the hinge region, a commonly exploited hotspot in kinase–ligand interactions. Additional π-alkyl interactions were formed with Ala72 (β3 strand), Lys74 (catalytic lysine on β3), and Cys193 (within the activation segment), as well as a secondary π-alkyl contact with Cys129, collectively contributing to the tight fit of apigenin in the kinase active site.

Interestingly, the orientation of pinostrobin in the binding pocket was flipped when compared to apigenin, possibly due to different substitution patterns. Docked pose analysis revealed multiple stabilizing interactions within the ATP-binding pocket of AAK1. Similar to apigenin, a hydrogen bond was established with residue Cys129, positioned in the hinge region connecting the N-terminal and C-terminal kinase lobes. Additionally, extensive hydrophobic interactions stabilized the ligand within the active site. Residues Leu52 and Val60, situated within the glycine-rich loop (P-loop), participated in π–sigma and π–alkyl interactions.

Furthermore, residues Leu183 and Ala72, located adjacent to the αC-helix and β-sheet regions, and Val104 and Cys193, positioned within the catalytic and activation loop regions, respectively, formed multiple π-alkyl contacts, effectively anchoring apigenin through interactions with its aromatic π-system. Additionally, a π-sulfur interaction involving Met126, a residue near the hinge region, further stabilized the ligand binding conformation (Figure 7). Collectively, these hydrophobic and polar interactions suggest that apigenin could effectively compete with ATP binding or stabilize the kinase in an inactive conformation, highlighting its therapeutic potential in modulating neuropathic pain through AAK1 inhibition.

## 4. Discussion

The investigated plant extracts demonstrated significant antihyperalgesic and anti-inflammatory activity in a rat model of DN. Among them, *Morus alba* showed the most consistent therapeutic profile, supporting its potential role as a strategy for managing DN.

In our study, DN was induced by a single i.p. injection of alloxan (130 mg/kg), a widely used model for studying diabetes and its complications due to its low cost, simplicity, and reproducibility [68,69]. The resulting hyperglycemia, caused by selective β-cell toxicity, leads to increased oxidative stress, a key factor in the development of pain hypersensitivity [70]. Therefore, untreated diabetic rats exhibited significantly heightened responses to thermal and mechanical stimuli over the 15 days, confirming the onset of hyperalgesia.

By day 15, none of the treatments restored normoglycemia; however, all four extracts significantly lowered blood glucose levels compared to the D group. MA had the most substantial effect (–45.7%), consistent with reports on its inhibition of α-glucosidase and improvement of insulin sensitivity [71]. For AAE, while direct hypoglycemic data are limited, related studies on angelana, a bioactive polysaccharide isolated from *Angelica gigas,* have shown its ability to prevent autoimmune diabetes in non-obese diabetic mice by reducing pancreatic lymphocyte infiltration and preserving β-cell function. This immunomodulatory capacity could partially underlie the glycemic improvements observed in our study [72]. PI also demonstrated significant hypoglycemic effects, aligning with earlier findings in streptozotocin-induced diabetic mice, by reducing fasting glucose levels, improving glucose tolerance, and normalizing the lipid profile [73]. VOE’s hypoglycemic effect may be related to the activation of PPARγ (peroxisome proliferator-activated receptor gamma) and the upregulation of adiponectin, thereby improving insulin sensitivity [74].

In behavioral testing, all plant extracts significantly reduced pain sensitivity across the hot plate, tail immersion, von Frey, and Randall–Selitto tests. The MAE exhibited the most pronounced effect, followed by VOE in thermal hypersensitivity and AAE in mechanical hypersensitivity tests. In heat and cold pain models, MAE and VOE performed comparably to gabapentin.

The MAE’s antihyperalgesic activity, as evidenced in our study, is supported by its rich profile of bioactive flavonoids, previously characterized in our phytochemical analysis. In a previous study, we identified high concentrations of isorhamnetin (approximately 61,936 µg/g) and quercetin (approximately 26,357 µg/g) in the MA extract [57]. These compounds are known for their antioxidant, anti-inflammatory, and neuroprotective activities. Quercetin has been shown to alleviate DN by activating the AMPK (AMP-activated protein kinase)/PGC-1α (peroxisome proliferator-activated receptor gamma coactivator-1α) pathway, improving mitochondrial function, increasing nerve conduction velocity, and preserving myelin integrity in streptozotocin-induced diabetic rats [75]. Similarly, isorhamnetin has demonstrated protective effects in high-fat diet/streptozotocin-induced diabetic mice by lowering blood glucose, suppressing pro-inflammatory cytokines (IL-1β, TNF-α), enhancing antioxidant defenses, and improving pain thresholds and nerve histology [76]. Other studies confirmed that mulberry flavonoids protect the sciatic nerve structure in diabetic rats [77]. These findings suggest that the phytoconstituents present in MAE may underlie the significant antihyperalgesic effects observed in our model of DN.

While MAE exhibited the most pronounced overall effect in our study, the antihyperalgesic activity observed for VOE was also significant. It may be attributed mainly to its rich content of bioactive phenolic compounds. Phytochemical analysis of VOE revealed high concentrations of chlorogenic acid (11,714 µg/g), along with measurable levels of *p*-coumaric acid and rutin [57]. Chlorogenic acid exhibits notable antinociceptive effects in various neuropathic pain models, primarily through its antioxidant and anti-inflammatory properties. It reduces mechanical and cold hyperalgesia by suppressing pro-inflammatory mediators such as TNF-α, nitric oxide, and interleukins, and by scavenging reactive oxygen species, which are key contributors to peripheral nerve injury. Chlorogenic acid also modulates neuronal excitability by enhancing GABA_A_ receptor activity in the spinal cord and regulating ion channels, such as Kv1.4 and acid-sensing ion channels [78]. Rutin, another compound identified in VOE, has also demonstrated strong neuroprotective and anti-diabetic effects. In streptozotocin-induced diabetic rats, rutin administration significantly alleviated mechanical and thermal hyperalgesia, restored antioxidant capacity, and reduced caspase-3 expression in dorsal root ganglia. These effects have been linked to the activation of the nuclear factor erythroid 2 (Nrf2) signaling pathway, which enhances cellular defenses against oxidative stress and promotes neuronal integrity [79]. Furthermore, *p*-coumaric acid has been shown to alleviate thermal and mechanical hyperalgesia and reduce oxidative stress in the sciatic nerve in a chronic constriction injury model, likely through the restoration of glutathione levels, which reinforces its potential contribution to the observed effects [80]. Additionally, VO extracts can enhance adiponectin via PPARγ, potentially contributing to reduced systemic inflammation [74].

The AAE’s antihyperalgesic effects are attributed to its high ferulic acid content (6350.87 µg/g), as determined by our phytochemical analysis [57]. Ferulic acid is widely recognized for its potent antioxidant and anti-inflammatory activities, and it has demonstrated efficacy in reducing pain behaviors and oxidative stress in experimental models of neuropathic pain. Its ability to scavenge free radicals and modulate inflammatory pathways supports the therapeutic relevance of AAE [81,82]. The extract also contains chlorogenic acid and rutin, compounds discussed earlier in connection with VOE.

The PIE had more moderate effects compared to MAE and VOE; however, its activity was consistent across behavioral tests. Phytochemical profiling of PIE revealed significant levels of flavonoids with well-known neuroprotective properties: isorhamnetin (1885 µg/g), *p*-coumaric acid (3155 µg/g), and quercetin (5660 µg/g) [57]. Their synergistic action may underlie the beneficial effects of PIE extract in DN.

Gabapentin, widely recommended as a first-line treatment for neuropathic pain, especially in post-herpetic neuralgia and DN [48], was the positive control in our study. Previous preclinical research demonstrated its antihyperalgesic properties [50,83]. In our study, gabapentin effectively reversed thermal and mechanical hypersensitivity, as highlighted by improvements in hot plate, tail withdrawal, von Frey, and Randall-Sellitto test responses in alloxan-induced DN in rats.

Biochemical analysis further supports the anti-inflammatory potential of the extracts. In our study, alloxan-induced DN significantly elevated TNF-α and IL-6 levels in brain and liver tissues, consistent with findings from previous studies [50]. Treatment with the investigated medicinal plant extracts, particularly AAE and MAE, resulted in substantial reductions in both cytokines. The AAE showed the most significant decrease in brain TNF-α (–64.25%), while MAE had the most considerable effect in liver tissue (–71.4%). Similarly, IL-6 levels were most reduced by MAE in the liver (–60.18%) and by AAE in the brain (–54.34%), with effects comparable to gabapentin. These findings suggest that behavioral improvements may be partially mediated by suppression of neuro-inflammatory pathways.

The cytokine reductions in the AAE-treated group likely stem from its high ferulic acid content [57], which has been shown to decrease TNF-α and IL-6 in vincristine-induced models [84]. Additionally, the notable decrease in TNF-α levels following AA extract treatment may also be linked to its chlorogenic acid content [57]. Chlorogenic acid, also present in AAE, contributes to the inhibition of NF-κB (nuclear factor kappa-light-chain-enhancer of activated B cells), JNK (c-Jun N-terminal kinase), ERK (extracellular signal-regulated kinase), and p38 MAPK (mitogen-activated protein kinase) pathways, thereby reducing cytokine production [85,86,87,88].

The significant reduction in TNF-α and IL-6 levels observed in the MAE-treated group may be attributed to its high content of quercetin and isorhamnetin [57]. Quercetin has been shown to downregulate the TLR4 (Toll-like receptor 4)/MyD88 (myeloid differentiation primary response 88)/NF-κB signaling pathway, leading to decreased expression of pro-inflammatory cytokines such as TNF-α and IL-6 in a rat model of DN [89]. Additionally, isorhamnetin significantly reduced these same cytokines in lipopolysaccharide-stimulated RAW264.7 macrophages and in vivo mouse models by inhibiting NF-κB activation [90].

Furthermore, the significant reduction in TNF-α and IL-6 levels observed in the VOE- and PIE-treated groups can be attributed to their bioactive profiles. The VOE contains high levels of chlorogenic acid, previously discussed for its anti-inflammatory effects through modulation of the NF-κB and MAPK pathways [57,87,88]. The PIE is rich in quercetin and isorhamnetin, flavonoids known to suppress the expression of pro-inflammatory cytokines through the inhibition of the NF-κB signaling cascade [57,89,90]. Gabapentin also significantly reduced TNF-α and IL-6 levels in both tissues, consistent with previous studies showing its ability to attenuate neuroinflammation [48,50].

Overall, MAE emerged as the most effective candidate, with significant antihyperalgesic and anti-inflammatory effects, particularly in reducing IL-6 levels. The extracts of AA, PI, and VO exhibited varying degrees of activity, with relevant efficacy depending on the test. These results support the pharmacological rationale for selecting these extracts based on GABAergic and anti-inflammatory mechanisms and justify further investigation in longer-term and dose-optimized DN models.

The combined in silico target prediction and molecular docking analyses support a mechanistic rationale for the observed antihyperalgesic effects of the plant extracts in the alloxan-induced diabetic neuropathy model. Several phytochemicals, previously identified in the four plant extracts, were predicted to modulate targets critically involved in neuropathic pain pathways, including CaV2.2, GluA2, AAK1, and aldose reductase (AR). The Super-PRED predictions are limited by the availability of data and are based on structural similarity to known ligands, with a focus on human protein targets. Therefore, some suggested targets may not directly translate to the rat model or could be false positives. These predictions served as a preliminary guide for our docking studies and required experimental validation to confirm their accuracy. Notably, many flavonoids showed high interaction probabilities with CaV2.2 and GluA2, two well-established regulators of synaptic transmission and neuronal excitability in chronic pain conditions. The predicted engagement of AAK1, a kinase involved in clathrin-mediated endocytosis and pain signaling, by compounds such as apigenin, naringenin, and genistein, suggests a novel modulatory route for the intervention of neuropathic pain. AAK1 represents a novel target in pain research, since its pharmacological inhibition produces antinociceptive effects in neuropathic pain models, mechanistically linked to α_2_-adrenergic signaling rather than opioid pathways [91]. Furthermore, AAK1 knockout mice show attenuated persistent pain responses in formalin models, underscoring its potential role in modulating long-lasting nociceptive signals. As a kinase that regulates AP2-mediated clathrin-dependent endocytosis, AAK1 influences receptor trafficking and potentially synaptic receptor availability; its inhibition may therefore modulate neuronal excitability and pain signaling via endocytic pathways [92].

Moreover, the predicted inhibition of aldose reductase, a key enzyme in the polyol pathway involved in diabetic nerve injury, provides a metabolic perspective on the neuroprotective effects of flavonoids, such as isorhamnetin, kaempferol, and rutin. These observations collectively suggest a multi-targeted mechanism, where phytochemicals have the potential to affect both neuronal excitability and metabolic stress responses relevant to diabetic neuropathy.

Interestingly, some flavonoids have been shown to have AR inhibitory activity in previous studies. For instance, quercetin was identified as a potent AR inhibitor (IC_50_ ≈ 25 µg/mL). Moreover, isoflavones genistein and formononetin also inhibit AR (IC_50_ ~57.1 µM and 69.2 µM, respectively), as do daidzein (~151.9 µM) and chlorogenic acid, a potent inhibitor at 4.2 µM. Glycosides such as genistin (genistein glucoside), hyperoside, and rutin (quercetin glucosides) are also expected to act as AR inhibitors once metabolized to their aglycones [93,94]. Several phenolic acids can also inhibit the activity of AR. Gallic acid, *p*-coumaric acid, syringic acid, and trans-cinnamic acid all showed measurable AR inhibition (IC_50_ in the ~0.1–0.17 mg/mL range in rat lens assays) [93]. Caffeic and ferulic acids are also reported as AR inhibitors in dietary studies, though with moderate potency [94,95]. Catechin (especially galloylated derivatives) has also been characterized as an AR inhibitor [96]. These AR-inhibiting compounds are of interest for diabetic complications, as blocking AR can alleviate hyperglycemia-induced damage. Several flavonoids, such as quercetin and rutin, have shown protective effects against diabetic neuropathy in animal models, likely due in part to their AR inhibition and antioxidant activity [97].

Further insights were gained from docking simulations targeting the ATP-binding pocket of AAK1. To the best of our knowledge, no previous studies have investigated the inhibitory activity of the screened phytochemicals against AAK1. Interestingly, some of the natural compounds contained in our plant extracts, such as quercetin, apigenin, kaempferol, and galangin, are widely recognized for their ability to inhibit various protein kinases [98,99,100]. In our computational study, several flavonoids, including hyperoside, naringenin, and pinostrobin, exhibited substantial predicted binding affinities, with binding energies ranging from –8.7 to –9.8 kcal/mol and favorable ligand efficiencies. Visual inspection of docked complexes revealed that key interactions with residues from the hinge region (Cys129), β3 strand (Lys74, Ala72), glycine-rich loop (Leu52, Val60), and activation segment (Cys193) were consistently involved in ligand stabilization. For example, apigenin and pinostrobin established hydrogen bonds with Cys129 and displayed π-stacking and π-alkyl contacts with several hydrophobic residues, highlighting a conserved binding mode. The presence of π-sulfur interactions with Met126 further supports the anchoring of flavonoids within the catalytic cleft. These structural insights, combined with the strong docking scores, reinforce the hypothesis that AAK1 inhibition may significantly contribute to the analgesic properties of the tested plant extracts, supporting further investigation of flavonoid scaffolds as natural lead compounds in neuropathic pain therapy.

The present study has several limitations that should be acknowledged. Firstly, the 15-day treatment period, constrained by alloxan-induced systemic toxicity, limits the assessment of long-term intervention effects. Considering the chronic nature of diabetic neuropathy, the limited 15-day treatment period precludes definitive conclusions regarding the long-term efficacy and safety of the investigated extracts. Consequently, it remains undetermined whether prolonged administration would maintain the observed therapeutic effects, confer additional cumulative benefits, or potentially give rise to delayed adverse outcomes. Additionally, although significant reductions in pro-inflammatory cytokines (TNF-α and IL-6) were observed, the study did not assess whether these effects are sustained over extended periods of time.

The absence of direct histopathological and electrophysiological assessments also represents a limitation; such analyses would provide more substantial evidence of neuropathic pathophysiology and substantiate the interpretation of the observed behavioral and biochemical outcomes. In addition, although molecular docking analyses indicated possible mechanisms of action, the absence of in vitro or ex vivo validation highlights the need for future studies to substantiate these predictions.

Another limitation is the lack of a dose–response analysis. In this study, only a single dose (200 mg/kg) was tested for each medicinal plant extract, based on the previous literature. This approach limits the ability to determine the minimum effective dose, the optimal therapeutic range, or whether higher doses might yield superior or more consistent antihyperalgesic and anti-inflammatory effects, particularly for extracts such as PIE, which have shown comparatively lower efficacy. Future studies are therefore warranted to incorporate dose-ranging experiments to refine therapeutic levels and optimize both efficacy and safety profiles.

Finally, the exclusive use of gabapentin as a reference, although appropriate in preclinical neuropathic pain models, restricts the comparative scope. Incorporating additional standard agents such as duloxetine or pregabalin in future studies would provide greater translational relevance.

## 5. Conclusions

The present study revealed that, among all plant extracts that produced significant improvements, *Morus alba* extract exhibited the most substantial overall effect, both behaviorally and biochemically, with the most notable reduction observed for IL-6 levels. These outcomes are likely mediated by its high content of quercetin and isorhamnetin, as well as its interaction with the GABAergic system. *Angelica archangelica* extract also exhibited strong anti-inflammatory properties, primarily attributed to its high levels of ferulic and chlorogenic acids, with a particularly potent effect on reducing TNF-α. Gabapentin validated its role as a reference compound, producing comparable reductions in thermal and mechanical hypersensitivity, as well as pro-inflammatory cytokine levels. Overall, *Morus alba* extract emerges as the most promising phytotherapeutic candidate for managing DN pain, warranting further investigation in long-term models and dose–response studies. The in silico findings suggest that the identified phytochemicals from the plant extracts studied might exert their antihyperalgesic effects through multi-target interactions, with AAK1 emerging as a promising molecular target for further exploration.

## Figures and Tables

**Figure 1 cimb-47-00719-f001:**
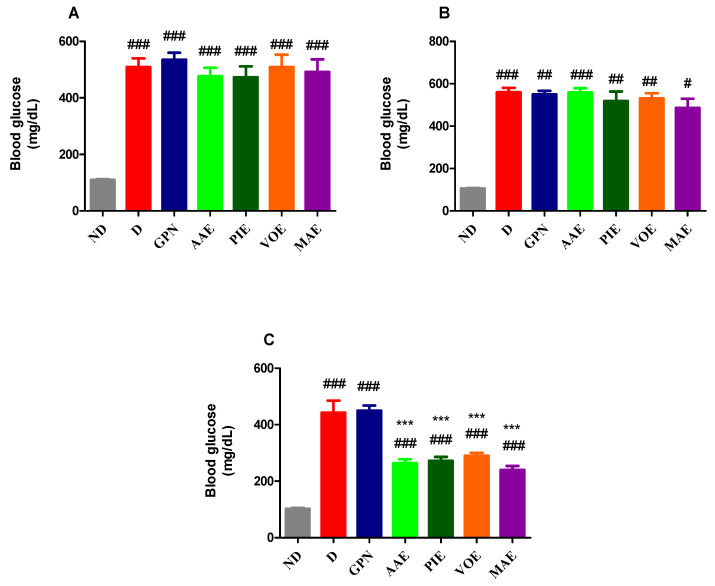
(**A**) Blood glucose levels were measured at baseline. (**B**) Blood glucose levels were measured on day 7. (**C**) Blood glucose levels were measured on day 15. ^#^ *p* < 0.05; ^##^ *p* < 0.01; ^###^ *p* < 0.001 vs. ND. *** *p* < 0.001 vs. D.

**Figure 2 cimb-47-00719-f002:**
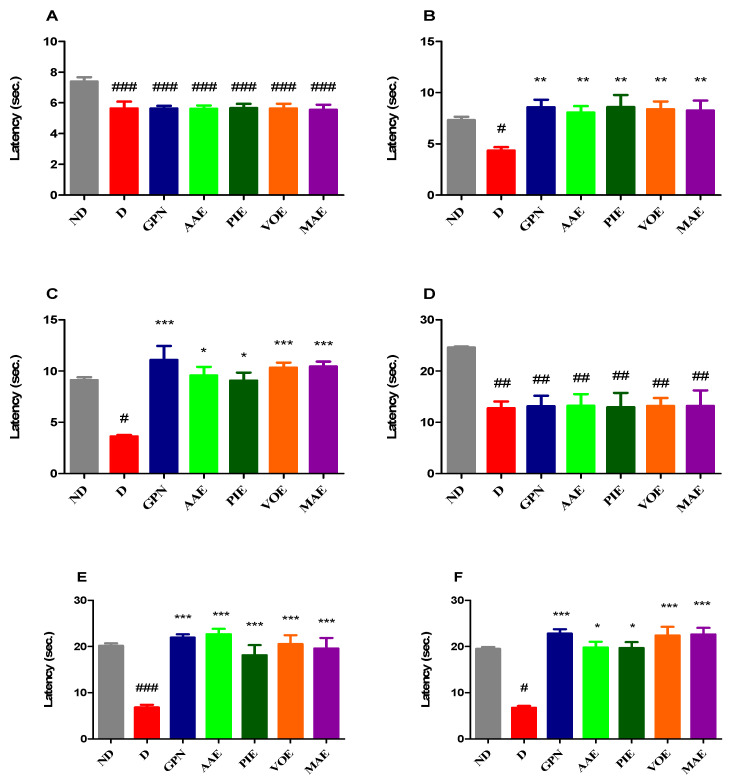
(**A**) Latency of the initial pain response in the hot plate test. (**B**) Latency of pain response in the hot plate test after 7 days. (**C**) Latency of pain response in the hot plate test after 14 days. (**D**) Baseline pain response latency in the tail withdrawal test. (**E**) Tail withdrawal test latency on day 7. (**F**) Tail withdrawal test latency on day 14. Data are presented as mean + S.E.M. ^#^ *p* < 0.05; ^##^ *p* < 0.01 ^###^ *p* < 0.001 vs. ND. * *p* < 0.05; ** *p* < 0.01; *** *p* < 0.001 vs. D.

**Figure 3 cimb-47-00719-f003:**
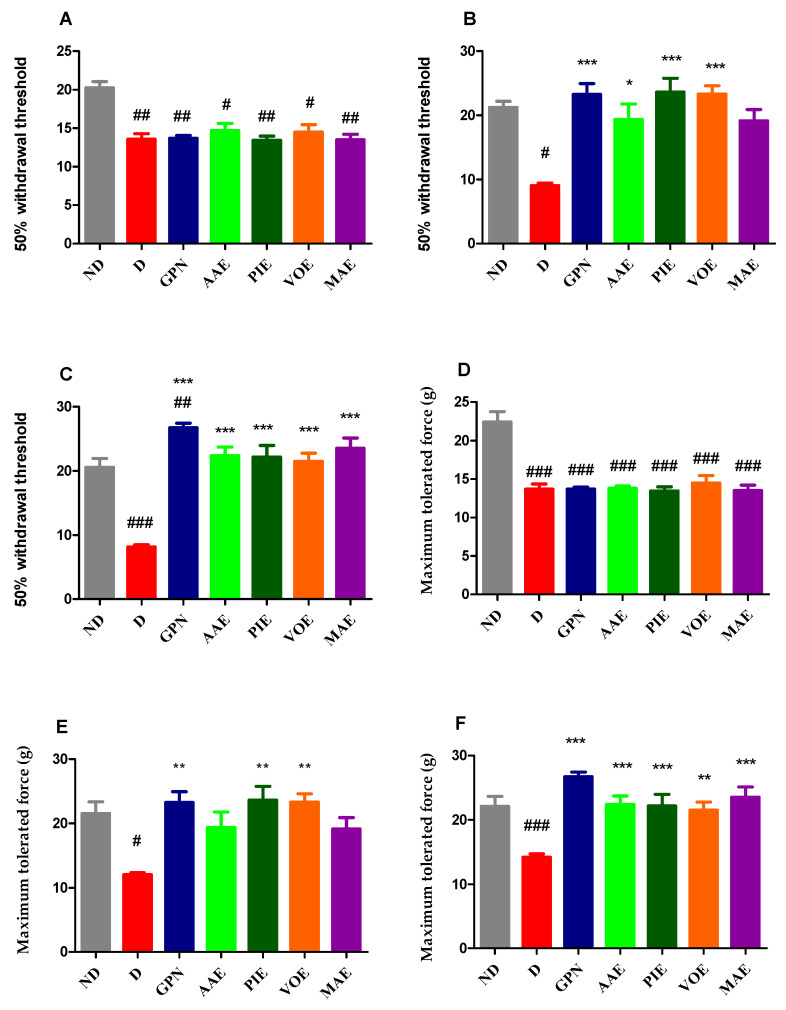
(**A**) 50% withdrawal threshold in the initial von Frey test. (**B**) 50% withdrawal threshold in the von Frey test on day 8. (**C**) 50% withdrawal threshold in the von Frey test on day 15. (**D**) Maximum tolerated force in the initial Randall–Sellito test. (**E**) Maximum tolerated force in the Randall–Sellito test on day 7. (**F**) Maximum tolerated force in the Randall–Sellito test on day 14. Values are expressed as mean + S.E.M. ^#^ *p* < 0.05;^##^ *p* < 0.01; ^###^ *p* < 0.001 vs. ND. * *p* < 0.05; ** *p* < 0.01; *** *p* < 0.001 vs. D.

**Figure 4 cimb-47-00719-f004:**
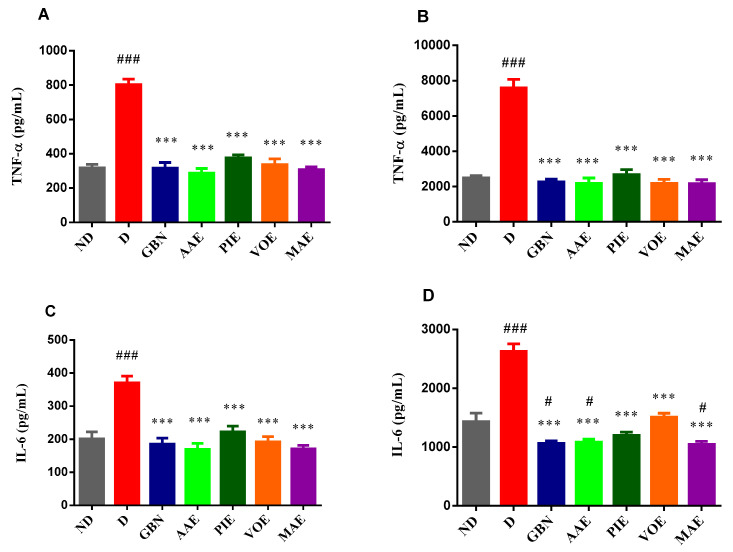
Pro-inflammatory cytokine levels in rat tissues after treatment. (**A**) TNF-α levels in brain tissue. (**B**) TNF-α levels in liver tissue. (**C**) IL-6 levels in brain tissue. (**D**) IL-6 levels in liver tissue. Values are expressed as mean + SEM ^#^ *p* < 0.05, ^###^ *p* < 0.01 vs. ND group. *** *p* < 0.001 vs. D group.

**Figure 5 cimb-47-00719-f005:**
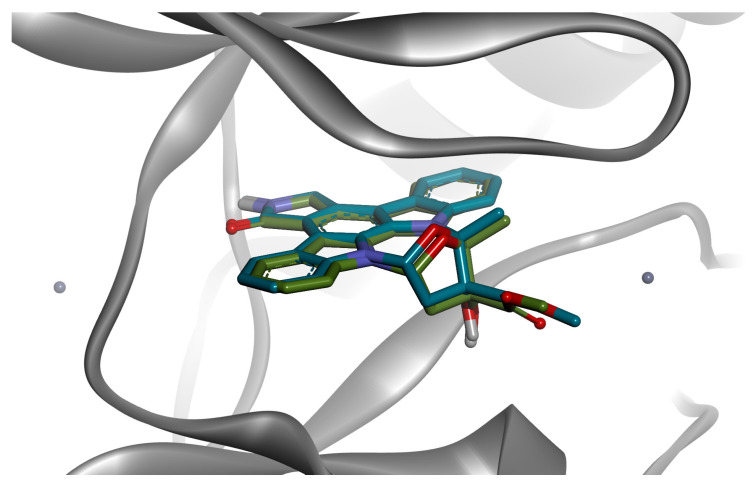
Superposition of the re-docked pose (blue) of the co-crystallized protein kinase inhibitor K-252A on the experimental conformation (green).

**Figure 6 cimb-47-00719-f006:**
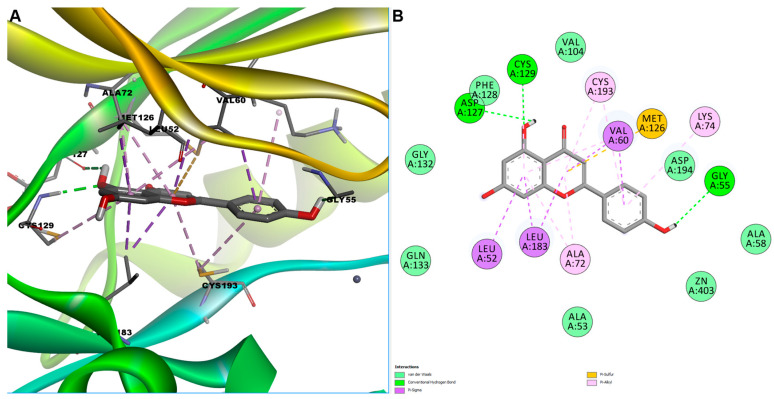
Predicted binding pose of apigenin in the AAK1 active site. (**A**) 3D conformation of predicted AAK1-apigenin complex; (**B**) 2D interactions diagram between apigenin and AAK1.

**Figure 7 cimb-47-00719-f007:**
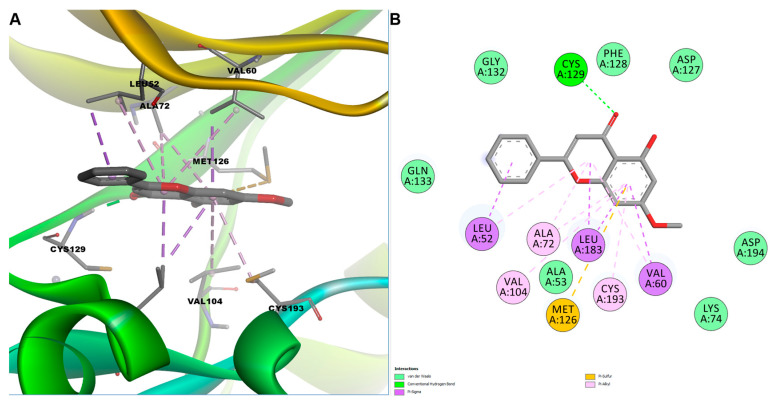
Predicted binding pose of pinostrobin in the AAK1 active site. (**A**) 3D conformation of predicted AAK1-pinostrobin complex; (**B**) 2D interactions diagram between pinostrobin and AAK1.

**Table 1 cimb-47-00719-t001:** Predicted neuropathy-associated targets for the phytoconstituents identified in the four plant extracts. AAK1 (Adaptor-associated kinase 1/AP2-associated protein kinase 1); A1R (Adenosine A_1_ receptor); AR (Aldose reductase); FAAH (Fatty acid amide hydrolase/anandamide amidohydrolase); CCR2 (C-C chemokine receptor 2); CatS (Cathepsin S); GluA2 (glutamate ionotropic receptor AMPA type subunit 2); MC4R (Melanocortin receptor 4); M1R (Muscarinic acetylcholine receptor M_1_); TP (Thromboxane A_2_ receptor); TRPA1 (Transient receptor potential ankyrin 1); CaV2.2 (Voltage-gated N-type calcium channel, α-1B subunit).

Compound	Target Names	Probabilities	Model Accuracies
Abscisic acid	CaV2.2	0.5029	0.9714
Apigenin	GluA2; TP; AAK1; CaV2.2; CatS	0.7366; 0.5245; 0.7149; 0.5516; 0.5376	0.8692; 0.9262; 0.831; 0.9714; 0.9560
Caffeic acid	CCR2; AAK1	0.5212; 0.5088	0.9857; 0.8310
Catechin	AAK1; CaV2.2	0.6417; 0.6223	0.831; 0.9714
Chlorogenic acid	GluA2; M1R; CaV2.2; CCR2; M1R	0.5414; 0.8814; 0.584; 0.5515; 0.8814	0.8692; 0.9423; 0.9714; 0.9856; 0.9423
Chrysin	TP; CaV2.2; M1R; CatS; M1R	0.5181; 0.6227; 0.5997; 0.5864; 0.5997	0.9262; 0.9714; 0.9423; 0.956; 0.9423
Cinnamic acid	M1R; M1R	0.6955; 0.6955	0.9423; 0.9423
Daidzein	GluA2; AAK1; CatS; CaV2.2; M1R; M1R	0.7268; 0.6016; 0.5966; 0.5599; 0.5094	0.8692; 0.8310; 0.956; 0.9714; 0.9423
Epicatechin gallate	TRPA1; FAAH; CaV2.2; M1R	0.5279; 0.5352; 0.5608; 0.6111	0.9217; 0.9753; 0.9714; 0.9423
Ferulic acid	GluA2; CaV2.2	0.5380; 0.5874	0.8692; 0.9714
Formononetin	GluA2; CaV2.2	0.8539; 0.7292	0.8692; 0.9714
Galangin	TP; M1R; AAK1; CaV2.2; CatS; M1R	0.565; 0.649; 0.6297; 0.5833; 0.5129; 0.649	0.9262; 0.9423; 0.831; 0.9714; 0.956; 0.9423
Gallic acid	M1R	0.5275	0.9423; 0.9423
Genistein	GluA2; TP; AAK1; CaV2.2; CatS	0.7874; 0.5564; 0.6801; 0.5701; 0.5596	0.8692; 0.9262; 0.831; 0.9714; 0.9560
Genistin	GluA2; TP; A1R; CaV2.2; FAAH; M1R	0.9175; 0.5174; 0.7768; 0.6142; 0.5615; 0.5261	0.8692; 0.9262; 0.9593; 0.9714; 0.9753; 0.9423
Glycitein	GluA2; CaV2.2; MC4R	0.7938; 0.7667; 0.513	0.8692; 0.9714; 0.9538
Hesperetin	GluA2; CaV2.2; AAK1; MC4R	0.7343; 0.8026; 0.5834; 0.5328	0.8692; 0.9714; 0.831; 0.9538
Hyperoside	GluA2; M1R; CaV2.2; A1R; M1R	0.8577; 0.6167; 0.5644; 0.5231; 0.6167	0.8692; 0.9423; 0.9714; 0.9593; 0.9423
Isorhamnetin	GluA2; AR; CaV2.2; AAK1	0.7553; 0.6451; 0.731; 0.6805	0.8692; 0.9238; 0.9714; 0.8310
Kaempferol	GluA2; TP; AR; AAK1; CaV2.2	0.6827; 0.5713; 0.5024; 0.8143; 0.5106	0.8692; 0.9262; 0.9238; 0.831; 0.9714
Naringenin	GluA2; AAK1; CaV2.2; CatS	0.6438; 0.738; 0.6124; 0.5395	0.8692; 0.831; 0.9714; 0.956
*p*-Coumaric acid	TP; M1R; CatS; AAK1; M1R	0.5049; 0.5597; 0.5493; 0.5370; 0.5597	0.9262; 0.9423; 0.956; 0.831; 0.9423
Pinocembrin	CaV2.2; M1R; AAK1; M1R	0.6498; 0.5695; 0.5141; 0.5695	0.9714; 0.9423; 0.831; 0.9423
Pinostrobin	CaV2.2; M1R	0.7959; 0.5779	0.9714; 0.9423
Quercetin	GluA2; TP; AAK1	0.6568; 0.5404; 0.8041	0.8692; 0.9262; 0.831
Rutin	GluA2; AR; A1R; CaV2.2	0.7631; 0.6648; 0.6359; 0.5362	0.8692; 0.9238; 0.9593; 0.9714
Syringic acid	CaV2.2	0.5062	0.9714

**Table 2 cimb-47-00719-t002:** Molecular docking results targeting the AAK1 ATP-binding site for screened phytochemicals.

Bioactive Phytochemicals	ΔG (kcal/mol)	LE
Hyperoside	–9.814	0.2974
Naringenin	–9.113	0.4556
Isorhamnetin	–8.955	0.3893
Pinostrobin	–8.946	0.4473
Quercetin	–8.945	0.4066
Apigenin	–8.876	0.4438
Kaempferol	–8.876	0.4227
Pinocembrin	–8.760	0.4611
Formononetin	–8.742	0.4163
Chrysin	–8.737	0.4598
Catechin	–8.727	0.4156
Genistin	–8.712	0.2810
Galangin	–8.700	0.4350
Glycitein	–8.549	0.4071
Hesperetin	–8.525	0.4060
Daidzein	–8.509	0.4478
Genistein	–8.496	0.4248
Rutin	–8.029	0.1867
Chlorogenic acid	–7.705	0.3082
Epicatechin gallate	–7.182	0.2244
Abscisic acid	–6.698	0.3525
Caffeic acid	–6.581	0.5062
Ferulic acid	–6.546	0.4676
*p*-Coumaric acid	–6.329	0.5274
Cinnamic acid	–6.327	0.5752
Syringic acid	–5.625	0.4018
Gallic acid	–5.537	0.4614

## Data Availability

The original contributions presented in this study are included in the article. Further inquiries can be directed to the corresponding authors.

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
