# Peer review of "Evaluation of the Antihyperalgesic Potential of Morus alba, Angelica archangelica, Valeriana officinalis, and Passiflora incarnata in Alloxan-Induced Diabetic Neuropathy in Rats"

_cimb, 2025, doi:10.3390/cimb47090719_

Round 1
Reviewer 1 Report
Comments and Suggestions for Authors
This is important and a somewhat underexploited area—diabetic neuropathy (DN), where some drug therapeutic options exist, but which are not effortless or satisfying to treat. Four medicinal plants (Morus alba, Angelica archangelica, Valeriana officinalis, and Passiflora incarnata) with GABAergic, antioxidant, and anti-inflammatory properties are focused on a novel approach aimed at multiple targets. Given the phytochemical variability and the traditional use of Morus alba in ethnomedicine, inclusion of it as a primary candidate for antihyperalgesic potential is particularly relevant. This multi-level assessment (behavioral nociception assays, biochemical cytokine profiling, and computational target prediction followed by molecular docking) ensures good triangulation of evidence. A battery of four pain assays (hot-plate, tail-withdrawal, von Frey, and Randall–Selitto) provides a comprehensive functional readout allowing the quantification of antihyperalgesic effects in thermal, mechanical, and tactile hypersensitivity.
Some suggestions below for improvement:
1) The treatment only lasted for 15 days in a chronic disease like DN, and therefore the long-term efficacy safety is not known.
2) A lack of direct histopathological or electrophysiologic evidence for nerve damage/regeneration, which would support a neuropathic pathophysiology.
3) Doses of plant extracts are based on previous literature but do not study dose-response; the use of a single fixed dose limits our understanding about effective therapeutic levels.
4) Gabapentin is used as a control; however, a comparison with other routine DN therapeutics (duloxetine, etc.) could potentially be lacking.
5) These studies identify potential molecular docking targets (AAK1, etc.), although in vitro/ex vivo target engagement validation is absent.
Author Response
Dear Reviewer 1,
The authors greatly appreciate your time and valuable review report, which aims to improve the quality of the present manuscript. They responded point by point to all comments and marked them in the revised manuscript using the track changes feature. Moreover, they revised the English Language using Grammarly 2025 v. Premium. Please see the attachment.

Reviewer 2 Report
Comments and Suggestions for Authors
The manuscript entitled "Evaluation of the Antihyperalgesic Potential of Morus alba, Angelica archangelica, Valeriana officinalis, and Passiflora incarnata in Alloxan-Induced Diabetic Neuropathy in Rats" presents a well-designed study combining in vivo behavioral tests, biochemical assays, and in silico analyses to address a clinically relevant problem—diabetic neuropathy—using natural products. The work is timely, translational, and of potential interest to the readership.
However, several aspects need clarification, additional detail, and revision to improve rigor, reproducibility, and clarity.
The manuscript requires thorough language and grammar editing to improve fluency, precision, and academic tone.
Statistical reporting needs more detail (test type, post-hoc corrections, justification for non-parametric tests). Inconsistencies (e.g., ANOVA vs Kruskal-Wallis) should be addressed.
While the docking predictions are interesting, they remain hypothetical. The manuscript sometimes presents these results as if they confirm biological activity; stronger emphasis on their predictive (not confirmatory) nature is needed, along with more methodological detail and validation steps.
Only a single dose (200 mg/kg) per extract was tested, limiting conclusions about optimal dosing. The 15-day treatment period may also be too short to fully model chronic diabetic neuropathy; this limitation should be better contextualized in the discussion.
Some figure references are out of order, captions contain formatting artifacts, and certain data are described before the corresponding figures appear. Figures should be re-embedded cleanly.
The introduction should provide a stronger justification for choosing these four plant species, supported by relevant literature.
Specific Comments and Suggestions
Abstract
- Dose justification – In “oral administration of the plant extracts (200 mg/kg/day)… gabapentin (100 mg/kg/day),” briefly note the rationale:
“(doses selected based on prior efficacy in rodent neuropathy models).”
- Results overload – Too many numerical results are presented. Focus on the most important findings and avoid overloading the reader.
Materials and Methods
- Extraction protocol (Lines 127–129) – Clarify why different ethanol concentrations were used for the plant materials, as this affects phytochemical profiles.
- Freeze-drying (Line 132) – Clarify whether –58 °C refers to shelf temperature or sample temperature.
- Blood glucose units (Lines 187–188) – Consider adding mmol/L alongside mg/dL, or justify exclusive use of mg/dL.
- Pain test terminology (Lines 204–205) – Use the standard term “tail-flick” instead of “tail-withdrawal” throughout.
- Target prediction (Lines 250–257) – Acknowledge limitations of the Super-PRED server and specify whether human or rat protein targets were used. Example:
“Super-PRED utilizes machine learning based on chemical similarity to known drugs; however, predictions require experimental validation due to potential false positives.”
- Docking validation (Lines 550–556) – Specify the docking software (e.g., AutoDock Vina), validate by re-docking co-crystallized ligands and reporting RMSD, and compare binding affinities to known inhibitors.
Results
- Statistical choice (Lines 319–327) – Justify use of Kruskal-Wallis if data are normally distributed; otherwise, consider ANOVA with appropriate post-hoc tests.
- Fold-change reporting (Lines 349, 374–375) – Avoid fold-change without a baseline reference; report % change compared to the diabetic control group and define “fold increase” in Methods.
- Figure formatting (Figure 4) – Remove artifacts, ensure clean embedding, and update caption to clearly indicate tissues, units, and statistical significance.
Discussion
- Opening paragraph (Lines 557–559) – Start with the key findings rather than a general statement.
- Mechanistic insight (Line 575–576) – Link cytokine reduction to specific phytochemicals (e.g., rutin, quercetin) and known mechanisms (NF-κB inhibition).
- AAK1 target (Lines 755–757) – Expand on the novelty of AAK1 as a target in pain research and provide mechanistic context (e.g., regulation of opioid receptor endocytosis and Nav1.7 channel trafficking).
References
- Update older citations (e.g., Chu et al. 2006; Kikuchi et al. 2010) with recent reviews such as:
Ansari P, Khan JT, Chowdhury S, Reberio AD, Kumar S, Seidel V, Flatt PR. Plant-based diets and phytochemicals in the management of diabetes mellitus and prevention of its complications: A review. Nutrients. 2024;16(21):3709.
Recommendation: Major Revisions
Author Response
Dear Reviewer 2,
The authors greatly appreciate your time and valuable review report, which aims to improve the quality of the present manuscript. The authors responded point by point to all comments and marked them in the revised manuscript using the track changes feature, and would be pleased to know that they succeeded. Moreover, they revised all figures and corrected the English language using Grammarly 2025, v. Premium. Please see the attachment.

Round 2
Reviewer 1 Report
Comments and Suggestions for Authors
The authors have made significant efforts to revise the manuscript in response to the reviewers' comments. The changes are substantial and address the concerns raised. I recommend the revised version for acceptance.
Author Response
C1. The authors have made significant efforts to revise the manuscript in response to the reviewers' comments. The changes are substantial and address the concerns raised. I recommend the revised version for acceptance.
C3. The authors would like to express their gratitude to Reviewer 1 for their time, attention, and the high professionalism of their accurate and valuable comments. They are also grateful for Reviewer 2's recommendation.
Reviewer 2 Report
Comments and Suggestions for Authors
Dear Editor,
I have carefully checked the revised version of the manuscript submitted to CIMB (ID: cimb-3820685), entitled “Evaluation of the Antihyperalgesic Potential of Morus alba, Angelica archangelica, Valeriana officinalis, and Passiflora incarnata in Alloxan-Induced Diabetic Neuropathy in Rats.”
The revisions adequately address the concerns raised, and the scientific content is solid and well-presented. I find the study suitable for publication. However, I recommend that the manuscript undergo English language proofreading by the editorial office to further improve readability and clarity.
Therefore, I suggest acceptance of the paper in its current form, subject only to minor language polishing.
Best regards,
Author Response
C1. I have carefully checked the revised version of the manuscript submitted to CIMB (ID: cimb-3820685), entitled “Evaluation of the Antihyperalgesic Potential of Morus alba, Angelica archangelica, Valeriana officinalis, and Passiflora incarnata in Alloxan-Induced Diabetic Neuropathy in Rats.”
R1. The authors would like to express their gratitude to Reviewer 2 for their time, attention, and the high professionalism of their accurate and valuable comments.
C2. The revisions adequately address the concerns raised, and the scientific content is solid and well-presented. I find the study suitable for publication.
R2. The authors are grateful for Reviewer 2's appreciation.
C3. Therefore, I suggest acceptance of the paper in its current form, subject only to minor language polishing.
R3. The authors thank Reviewer 2 again for's recommendations.
The entire manuscript was rechecked using Grammarly 2025 v. Premium, and all misprints were corrected. Additionally, the requested minor language polishing was performed.